# Leaving the barn door open for Clever Hans: Simple features predict LLM benchmark answers

## Abstract

The integrity of AI benchmarks is fundamental to accurately assess the capabilities of AI systems. The internal validity of these benchmarks—i.e., making sure they are free from confounding factors—is crucial for ensuring that they are measuring what they are designed to measure. In this paper, we explore a key issue related to internal validity: the possibility that AI systems can solve benchmarks in unintended ways, bypassing the capability being tested. This phenomenon, widely known in human and animal experiments, is often referred to as the 'Clever Hans' effect, where tasks are solved using spurious cues, often involving much simpler processes than those putatively assessed. Previous research suggests that language models can exhibit this behaviour as well. In several older Natural Language Processing (NLP) benchmarks, individual $n$-grams like "not" have been found to be highly predictive of the correct labels, and supervised NLP models have been shown to exploit these patterns. In this work, we investigate the extent to which simple $n$-grams extracted from benchmark instances can be combined to predict labels in modern multiple-choice benchmarks designed for LLMs, and whether LLMs might be using such $n$-gram patterns to solve these benchmarks. We show how simple classifiers trained on these $n$-grams can achieve high scores on several benchmarks, despite lacking the capabilities being tested. Additionally, we provide evidence that some modern LLMs might be using these superficial patterns to solve benchmarks. This suggests that the internal validity of these benchmarks may be compromised and caution should be exercised when interpreting LLM performance results on them.

## 1 Introduction

With the rise of large language models (LLMs), numerous benchmarks have been developed to evaluate different capabilities of these models and to rank their performance (see the collections HELM (Liang et al., 2022), BIGBench (Srivastava et al., 2022), LogiGLUE (Luo et al., 2023), CALM-bench (Dalal et al., 2023) and GLoRE (Teng et al., 2023) for examples of capabilities being tested). Ensuring the validity of these benchmarks is essential. One important aspect of validity is internal validity: the benchmarks should be free from confounding factors (Liao et al., 2021; Frank et al., 2023; Rutar et al., 2024). In other words, the behaviour and performance observed in a benchmark should be explainable by the specific capability (or lack thereof) being tested, not by other unrelated capabilities, systematic errors, or biases in the benchmark data.

A benchmark's internal validity can be compromised in several ways, including annotation errors and disagreement (Ivanova, 2023), selection bias (Gururangan et al., 2018), implementation variations (Liao et al., 2021; Biderman et al., 2024), limited linguistic diversity (McIntosh et al., 2024), and insufficient variability in both task-relevant and irrelevant features (Rutar et al., 2024; Yang et al., 2023). Additionally, a lack of prerequisite capabilities in the system to be tested (Rutar et al., 2024; Hu & Frank, 2024) and train-test contamination (Liao et al., 2021; Dominguez-Olmedo et al., 2024) can also impact validity. Some of these systematic issues may introduce "shortcuts" in benchmarks, which makes it possible for subjects to successfully complete tasks *without* using the specific capabilities the benchmarks are designed to test. In this paper, we investigate whether such shortcuts persist in modern LLM benchmarks and, if so, whether LLMs are exploiting them.

The phenomenon of using shortcuts or cues to solve tasks is known as the Clever Hans effect and is well documented in human and animal experimentation. It occurs when, for example, a participant's behaviour is influenced by unintended cues from the experimenter, leading to responses based on those cues rather than the task itself (Frank et al., 2023; MaBouDi et al., 2021). To control for such effects, techniques such as manipulation checks, stimulus randomisation and counterbalancing are used to detect and mitigate these issues.

In AI, the Clever Hans effect has also been widely recognised (Sturm, 2014; Hernandez-Orallo, 2019). In particular, reinforcement learning agents are known for solving tasks in unintended ways (Krakovna et al., 2020). One of the main goals of explainable AI is to understand the information AI models use to solve tasks and ensure they are relying on the right cues rather than exploiting spurious correlations (Ribeiro et al., 2016). In NLP benchmarks, previous research has shown that simple features extracted from prompts can correlate with labels, despite not being related to the capability being tested (Gururangan et al., 2018; Friedman et al., 2022; Gardner et al., 2021). Moreover, supervised language models have been found to exploit these features to solve tasks. For example, Kavumba et al. (2019) showed that an individual unigram "a" has been exploited by BERT to choose the correct alternative on COPA dataset. Most of the works in this space has focused on traditional NLP models. Fewer studies (e.g. Tu et al., 2020; Kavumba et al., 2022; Du et al., 2023) have investigated LLMs, and these have primarily considered individual $n$-grams or similarly simple features.

LLMs, however, have sufficient capacity to learn and rely on patterns involving multiple simple features. In this paper, we explore the extent to which current LLM benchmarks can be solved using *a combination of* such simple features. Specifically, we train logistic regression classifiers on unigrams and bigrams extracted from prompts to predict the correct labels in multiple-choice benchmarks where the choices are identical throughout benchmark samples. These classifiers lack the capacity to develop the capabilities the benchmarks are designed to test, meaning that any labels successfully predicted using these classifiers indicate that the examples may be solved without invoking the capability of interest.

We find that, in several benchmarks, our classifiers can accurately predict ground truth labels, raising concerns about internal validity. Moreover, recent work has highlighted the sensitivity of LLM performance to small perturbations in benchmark data suggesting that LLMs rely on shallow, surface level patterns to solve tasks (Alzahrani et al., 2024). The compromised internal validity of benchmarks may therefore allow LLMs to exploit these superficial patterns to solve tasks, rather than engaging the capabilities of interest. Indeed, our further analysis shows that some LLM families may leverage such $n$-gram associations to solve benchmark tasks, suggesting that they may *not* be using the intended capabilities to solve those tasks. However, our findings are not conclusive and warrant further investigation. In particular, we find no evidence of reliance on the spurious features we investigate for some other LLMs, but these LLMs may rely on other features we have not considered.

## 2 RELATED WORK

### 2.1 BENCHMARK VALIDITY

Several factors contribute to the validity of measuring an AI system's performance on a specific benchmark. One key factor is "construct validity", which refers to how well the benchmark actually measures the construct that is intended to be measure. A prerequisite for construct validity is "internal validity", which reflects the extent to which the observed outcomes can be directly attributed to the variables or factors under investigation, without interference from confounding variables or biases. Other forms of validity include ecological and external validity, which refer to the extent to which the benchmark tasks reflect real-world tasks and how the measurement results generalise to real-world performance (Davis, 2023; Ivanova, 2023; Elliot et al., 2024; Liao et al., 2021).

Several papers review issues related to benchmark validity. For instance, Liao et al. (2021) overview problems with the benchmarking paradigm of supervised ML systems, while Bowman & Dahl (2021) outline criteria NLP benchmarks should meet, some of which contribute to construct and internal validity. Subramonian et al. (2023) instead conduct a survey of AI researchers to develop a taxonomy of validity issues. McIntosh et al. (2024) identify a set of inadequacies in LLM bench-

marks, some related to validity, while others are broader, such as security issues and the failure to capture cultural diversity. In contrast, Biderman et al. (2024) focuses primarily on reproducibility issues, which can also impact benchmark validity. Finally, Momennejad et al. (2023) and Rutar et al. (2024) provide guidance on building evaluation frameworks to measure whether AI systems (LLMs in the former case and embodied agents in the latter) possess capabilities that are robust to various perturbations and experimental conditions, thereby ensuring both construct and internal validity.

## 2.2 Spurious correlations in NLP benchmarks

The presence of spurious correlations in NLP benchmarks has been demonstrated by several studies, with some showing that these correlations affect the performance of NLP models. For example, Gururangan et al. (2018) identified individual words associated with specific class labels in NLP benchmarks and trained a model on truncated prompts achieving performance above chance, despite removing necessary elements for determining the correct answer. Moreover, they showed that models trained on such datasets performed better on instances that could be successfully classified using the truncated prompts. Poliak et al. (2018), Si et al. (2019), Sugawara et al. (2020) and Kavumba et al. (2019) performed similar experiments on different benchmarks and found comparable results. Niven & Kao (2019) showed that individual unigrams and bigrams were related to label values for specific benchmarks and crafted instances by adversarially manipulating the $n$-grams such that NLP models achieved poor accuracy on them. For example, they showed that BERT was able to exploit the presence of the unigram "not" and bigrams like "will not" to correctly answer instances in the ARCT dataset (Habernal et al., 2018). In contrast, our approach identifies patterns involving multiple unigrams and bigrams at the same time.

In the context of LLMs, Tu et al. (2020) found evidence that pre-trained models finetuned on datasets with spurious correlations can learn to exploit these correlations. Similarly, Du et al. (2023) argued that the supervised fine-tuning step in the LLM development pipeline makes them vulnerable to exploiting spurious correlations. Kavumba et al. (2022) conducted experiments where they shuffled the words in the possible choices and truncated the prompts, following the approach of Gururangan et al. (2018), to demonstrate that LLMs exploit superficial cues; for instance, they found that RoBERTa exploits superficial cues like the word "not". Instead, Yan et al. (2024) shows how LLMs can learn to rely on spurious features present in task examples included in the context window as part of few-shot prompting. In contrast, we do not use few-shot examples, thus studying spurious correlations LLMs learnt during training.

Finally, more advanced methods for identifying spurious features have been proposed. Friedman et al. (2022) used grammar induction techniques to define features as sub-trees within a probabilistic grammar. Similarly, Gardner et al. (2021) introduced a statistical test to determine whether a spurious correlation arises from bias in the data generation process or is instead randomly introduced. One limitation of this approach however is that it only considers the correlation between individual features and the labels.

Here, we identify spurious correlations by training simple logistic regression classifiers to predict labels using only unigrams and bigrams extracted from the prompts. This approach allows us to explore whether combinations of these features are predictive of the correct response labels (e.g., "true" vs. "false") while remaining computationally lightweight, meaning that the approach is scalable to large datasets (unlike, for example, grammar induction techniques).

## 3 Methods

We aim to investigate (1) the extent to which we can predict labels on multiple-choice LLM benchmarks using combinations of simple unigrams and bigrams, and (2) whether LLMs rely on these features to succeed in these tasks. To this end, we evaluate the predictive power of different classifiers across benchmark datasets to correctly classify labels based on these $n$-gram features. Furthermore, we analyse the performance of LLMs on dataset instances that were successfully and unsuccessfully predicted by the $n$-gram models to determine if LLMs rely on these $n$-gram patterns to solve benchmark tasks.

| Dataset | # tested instances | # choices | Collection |
|---|---|---|---|
| Fantasy Reasoning | 197 | 2 | BIG-Bench |
| NeuBAROCO (Ando et al., 2023) | 363 | 3 | - |
| Moral Permissibility | 338 | 2 | BIG-Bench |
| Causal Judgment | 184 | 2 | BIG-Bench |
| Metaphor Boolean | 676 | 2 | BIG-Bench |
| Commonsense QA 2.0 (Talmor et al., 2021) | 2537 | 2 | - |
| SpaceNLI (Abzianidze et al., 2023) | 1600 | 3 | - |
| ANLI (Nie et al., 2019) | 3196 | 3 | - |
| ART (Collier et al., 2022) | 364 | 2 | - |
| WANLI (Liu et al., 2022) | 1000 | 3 | - |
| bAbI Task 16 | 1000 | 4 | BIG-Bench |
| Formal Fallacies Syllogisms Negation | 1000 | 2 | BIG-Bench |
| Abercrombie | 95 | 5 | LegalBench |
| Corporate Lobbying | 3267 | 2 | LegalBench |
| Function of Decision Section | 367 | 7 | LegalBench |
| PROA | 95 | 2 | LegalBench |
| International Citizenship Questions | 1000 | 2 | LegalBench |
| CLadder (Jin et al., 2023) | 8917 | 2 | - |
| ProntoQA (Saparov & He, 2023) | 7200 | 2 | - |

Table 1: The number of tested instances, the number of possible choices, and the collection to which each dataset belongs. For datasets from LegalBench, instance-level LLM performance results are available from Liang et al. (2022). Likewise, instance-level results for CLadder (Jin et al., 2023) and ProntoQA (Saparov & He, 2023) are also publicly available. For the remaining datasets, we obtained the instance-level LLM performance results ourselves.

## 3.1 DATASETS AND LLMS

We conducted experiments on a diverse set of nineteen LLM benchmarks, covering a wide range of linguistic and reasoning tasks: causal reasoning, counterfactual analysis, moral judgment and decision making, different types of formal reasoning, metaphor understanding, commonsense reasoning, spatial reasoning, legal reasoning, and natural language inference. All datasets used are multiple-choice, with the same possible choices across all instances within each dataset. Details of datasets used can be found in Table 1. Some of these datasets are from BIG-Bench (Srivastava et al., 2022), while others are from LegalBench (Guha et al., 2023). Instance-level performance data was collected from forty-four LLMs across eleven model families. For the larger datasets, the LLMs were tested on a random subset of instances to reduce costs; the number of tested instances is indicated in Table 1. Note that not all LLMs were tested on every dataset. For more details on the LLMs included in our analyses and the datasets they were tested on, see Appendix A.

## 3.2 FEATURE EXTRACTION

We considered thirteen different feature vectors derived from the prompts of each benchmark. To generate these features, we extracted unigrams and bigrams at both the word level and the token level; for token-level features, we used the GPT-2 tokenizer to capture subword information. For each set of $n$-grams, we computed three types of feature representations:

1. **Term Frequency (TF)**: The raw count of each $n$-gram in the prompt.

2. **Term Frequency-Inverse Document Frequency (TF-IDF)**: A weight reflecting how important an $n$-gram is to a prompt in the context of the entire dataset.

3. **Presence**: A binary indicator where each feature is 1 if the $n$-gram occurs in the prompt and 0 otherwise.

This process resulted in twelve feature vectors: *(unigrams, unigrams + bigrams)* × *(word level, token level)* × *(TF, TF-IDF, Presence)*. Additionally, we extracted a *Readability and Diversity Metrics* vector, which includes textual features such as the Flesch Reading Ease Score, Gunning Fog Index,

SMOG Index, and Yule's K adapted from Moros Daval (2023). In total, we obtained thirteen feature vectors for our analysis.

### 3.3 PERFORMANCE EVALUATION

To assess the performance of our models, we used Cohen's $\kappa$ coefficient (McHugh, 2012), a statistic that measures the agreement between two raters (or, in our case, between the model's predictions and the true labels) while accounting for the agreement occurring by chance. This metric is particularly suitable in our context as the datasets differ in the number of possible labels (see Table 1), leading to varying values of chance level performance. Cohen's $\kappa$ is calculated as:

$$\kappa = \frac{P_o - P_e}{1 - P_e}$$

where $P_o$ is the observed accuracy (the proportion of correctly predicted labels), and $P_e$ is the expected accuracy if predictions were made purely by random guessing. The value of Cohen's $\kappa$ ranges from $-1$ to $1$, where a value of $1$ indicates that the model perfectly predicts the labels (well beyond chance level), a value of $0$ implies that the model's performance is no better than random guessing, and negative values suggest systematic disagreement between the model's predictions and the true labels.

### 3.4 EXPERIMENTAL SETUP

For each dataset, we partitioned the data into training, validation, and test splits. To explore whether $n$-grams are predictive of labels, we trained logistic regression classifiers on the training split using each of the thirteen feature vectors discussed in Section 3.2. We tested both L2 regularization (with regulation weight $\lambda = 1$) and L1 regularization (with $\lambda = 1$ and $\lambda = 10$). For each dataset and feature vector, we selected the hyperparameters that achieved the highest accuracy on the validation split, and we report the corresponding results on the test split (see Section 4.1).

Additionally, to investigate whether LLMs are relying on $n$-gram patterns identified by the logistic regression classifiers, we further stratified the test split into two subsets: instances whose labels were successfully predicted by the classifier using the best-performing features on the validation split, and those whose labels were not successfully predicted. We then analyse the performance of the LLMs on these two subsets across the various datasets, as detailed in Section 4.2.

In the following section, we present the results of our experiments and discuss the implications of our findings.

## 4 RESULTS

### 4.1 DO $n$-GRAMS PREDICT GROUND TRUTH LABELS?

We first investigate the extent to which logistic regression classifiers built on $n$-grams can predict the ground truth labels of various datasets. As mentioned above, we train the classifiers with different sets of hyperparameters on the training split of each dataset, select the configuration that leads to the best validation accuracy and evaluate the performance on the test split.

Figure 1 shows Cohen's $\kappa$ values for all feature vectors across all datasets. For some datasets, relatively high values of Cohen's $\kappa$ can be achieved, while for others the values are negative or close to 0. This indicates that the ground truth labels can be predicted using simple features for some datasets, whereas for others this is not the case, at least with $n$-gram features. One could conjecture that high performance with $n$-gram models indicates an easy dataset. However, this does not seem to be the case: the distribution of LLM performance across different datasets is similar regardless of whether $n$-gram model performance is high or low (Figure 4 in Appendix A). Moreover, using TF-IDF seems to generally reduce predictive performance; this is expected as the IDF correction downweights the importance of $n$-grams that appear frequently across the datasets, which are the ones that may be more strongly correlated with the labels. Conversely, TF and Presence yield mostly

| Dataset | Log Reg settings | Features |
|---|---|---|
| CLadder | L1, $\lambda = 1$ | 2-grams TF, word-level |
| ProntoQA | L1, $\lambda = 1$ | 2-grams TF, word-level |
| ANLI | L1, $\lambda = 1$ | Readability and Diversity Metrics |
| ART | L1, $\lambda = 10$ | Readability and Diversity Metrics |
| bAbI Task 16 | L1, $\lambda = 1$ | 2-grams TF, word-level |
| Causal Judgment | L1, $\lambda = 10$ | Readability and Diversity Metrics |
| Commonsense QA 2.0 | L1, $\lambda = 1$ | Readability and Diversity Metrics |
| Fantasy Reasoning | L1, $\lambda = 10$ | 1-grams TF, word-level |
| Formal Fallacies Syllogisms Negation | L2, $\lambda = 1$ | Readability and Diversity Metrics |
| Abercrombie | L2, $\lambda = 1$ | Readability and Diversity Metrics |
| Corporate Lobbying | L2, $\lambda = 1$ | 1-grams TF, word-level |
| Function of Decision Section | L2, $\lambda = 1$ | 2-grams TF, token-level |
| International Citizenship Questions | L1, $\lambda = 1$ | 1-grams TF, token-level |
| PROA | L2, $\lambda = 1$ | 1-grams TF, token-level |
| Metaphor Boolean | L2, $\lambda = 1$ | Readability and Diversity Metrics |
| Moral Permissibility | L2, $\lambda = 1$ | Readability and Diversity Metrics |
| NeuBAROCO | L1, $\lambda = 1$ | 2-grams TF, word-level |
| SpaceNLI | L1, $\lambda = 1$ | 2-grams TF, word-level |
| WANLI | L2, $\lambda = 1$ | Readability and Diversity Metrics |

Table 2: The feature vector and hyperparameter values for logistic regression that achieved the highest accuracy in predicting the ground truth on the validation split for each dataset.

identical results, likely because most of the $n$-grams useful for predicting labels tend to appear only once.

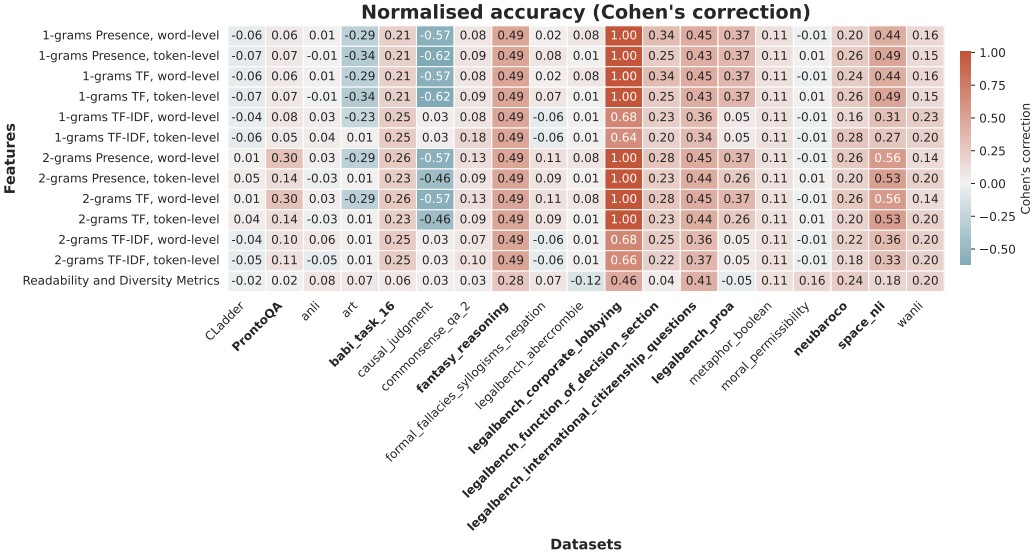

Figure 1: For each dataset and each feature vector, the best-performing classifier (i.e. the classifier with the highest Cohen's $\kappa$) on the validation split was selected. The heatmap shows the accuracy of these classifiers on the test split, measured in terms of Cohen's $\kappa$. Higher values indicate stronger predictability. The datasets for which the best set of features achieves $\kappa > 0.2$ are in bold.

In the following, for each dataset, we consider the feature vector that leads to the highest validation accuracy, as shown in Table 2.

## 4.2 Do LLMs rely on $n$-grams to succeed?

Here we analyse the performance of LLMs on instances that were successfully and unsuccessfully predicted by the $n$-gram classifiers. First, for each LLM and dataset, we compute the difference in performance between the two test subsets, and plot that against the overall test performance of the LLM on that dataset in Figure 2. The points in the plot are coloured according to the test performance of the best $n$-gram model.

The figure shows that the proportion of points where $n$-gram performance is high is the greatest in the upper right quadrant, which corresponds to LLM-dataset pairs with above-chance overall performance and a positive difference in performance between the two subsets. This suggests that some LLMs may be succeeding on specific datasets by relying on $n$-gram features. However, we note that there is also a smaller number points with high $n$-gram performance and low or close to $0$ difference between subsets, as well as points with a strongly positive difference between subsets but poor $n$-gram performance. This indicates that LLMs are not robustly relying on the identified patterns in $n$-grams and that statistical noise may also be contributing to positive differences in performance between subsets.

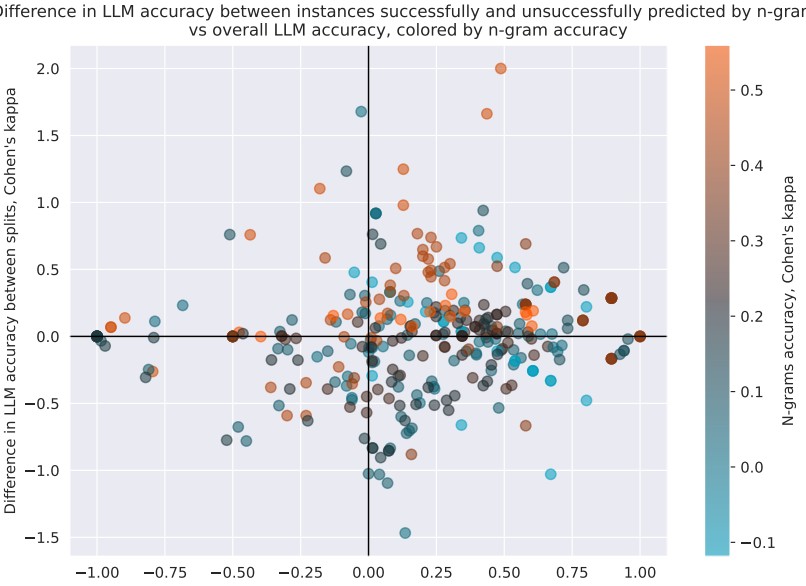

Figure 2: Overall LLM test accuracy and difference in accuracy between the two test split subsets. Each point represents a LLM tested on a dataset. The color scheme represents the test accuracy of the best $n$-gram based classifiers on those same datasets. The accuracy is reported using Cohen's $\kappa$, with higher values indicating stronger predictive power.

To further investigate whether LLMs may be relying on $n$-gram features, we conducted the following analysis. First, we discarded all datasets for which Cohen's $\kappa$ on the test split is less than $0.2$, which we consider the threshold above which a small but detectable agreement exists (Landis, 1977). The kept datasets are marked in bold in Figure 1. Then, in Figure 3, we plotted the performance distribution of each LLM on the two subsets of instances for the datasets on which they were tested. The figures shows that a number of LLMs perform better on the successfully predicted subset compared to the unsuccessfully predicted one. Moreover, we find that these LLMs are primarily from a few specific model families, in particular Meta, OpenAI and Mistral AI.

For each LLM family we conducted one-sided paired t-tests to test the hypothesis that the average LLM accuracy (on the datasets they were tested on) is higher on the subset of test instances whose labels were successfully predicted by the logistic regression classifiers than on the subset of instances where the labels were not successfully predicted. We obtained $p$-values from these t-tests and adjusted them for multiple comparisons using the Benjamini-Hochberg procedure (Benjamini

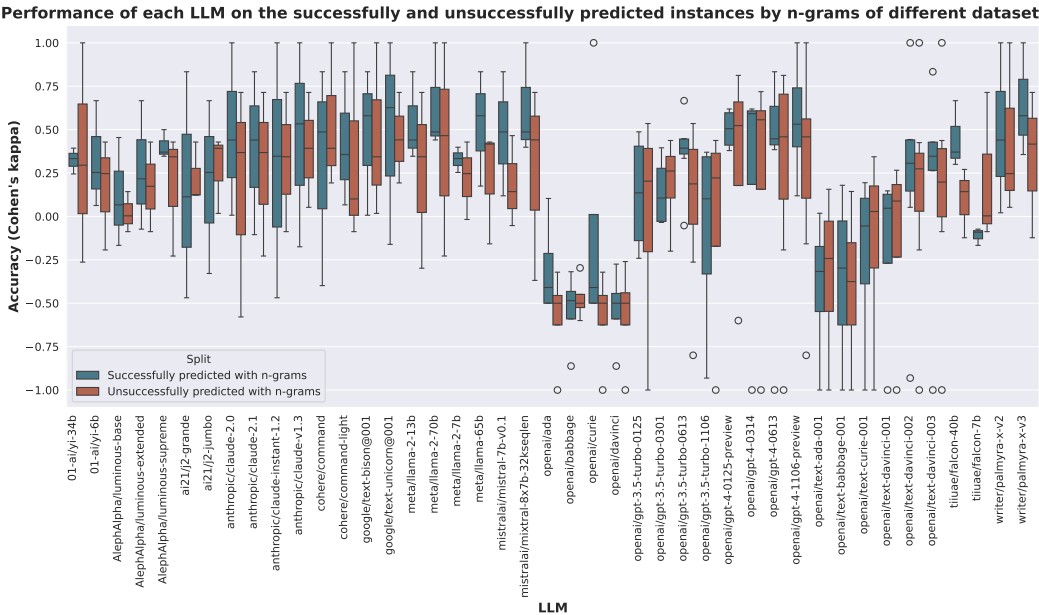

Figure 3: Accuracy of each LLM (measured using Cohen's $\kappa$; higher values indicate stronger predictability) on the two subsets of test instances. Each box represents the distribution of accuracy of the datasets tested with the considered LLM; notice not all datasets were tested with all LLMs (see Appendix A).

& Hochberg, 1995). The results are shown in Table 3. We find moderate evidence (adjusted $p$-values below 0.1) for the claim that models from OpenAI, Meta, Mistral AI and Writer are more likely to provide correct answers for instances that $n$-gram models also successfully predict. We find weaker evidence for Anthropic and Aleph Alpha (adjusted $p$-values below 0.2) and little to no evidence for other models. However, we note that many model families include only two LLMs (see Table 3) tested on just five datasets (see Table 4). To detect similar effect sizes as those found for model families like OpenAI, which included 18 LLMs tested on at least a dozen benchmarks, additional data may be required.

| LLM family | $p$-value | Adjusted $p$-value | Number of LLMs per family |
|---|---|---|---|
| Meta | 0.007342 | 0.059433 | 4 |
| OpenAI | 0.010806 | 0.059433 | 18 |
| Mistral AI | 0.027277 | 0.089484 | 2 |
| Writer | 0.032540 | 0.089484 | 2 |
| Anthropic | 0.071911 | 0.158205 | 4 |
| Aleph Alpha | 0.108315 | 0.198578 | 3 |
| 01-AI | 0.336185 | 0.528291 | 2 |
| Google | 0.393498 | 0.541060 | 2 |
| TII UAE | 0.474044 | 0.579387 | 2 |
| AI21 Labs | 0.678992 | 0.681576 | 2 |
| Cohere | 0.681576 | 0.681576 | 2 |

Table 3: LLM families, along with $p$-values and adjusted $p$-values obtained from one-sided paired t-tests testing the hypothesis that the average accuracy of LLMs belonging to a family (on the datasets they were tested on) is higher on the subset of test instances whose labels were successfully predicted by the logistic regression classifiers than on the subset of instances where the labels were not successfully predicted.

## 5 DISCUSSION

In this series of experiments, we demonstrate that simple unigram and bigram features extracted from benchmark prompts can be used to predict the correct answers for several multiple-choice LLM benchmarks. Furthermore, we provide evidence suggesting that at least some LLM families may be relying on these features to solve these benchmarks.

### 5.1 CONTROLLING FOR CLEVER HANS

Confounding factors are an issue in all forms of capability evaluation, potentially leading to misleading conclusions based on performance data. These confounds can be related to characteristics of the test participants—for example, older people tend to have encountered more words and thus have higher vocabulary, meaning that IQ tests premised on word-meaning knowledge will consistently attribute higher IQ to older participants (Verhaeghen, 2003))—or they can be related to features of the task itself. Within human behavioural research, it is common wisdom that such confounds are inevitable, and considerable effort goes into identifying, eliminating and controlling for them. In animal research, experimenters will often wear opaque goggles (Tsukasa et al., 2012; Mercado III et al., 2000) or use pre-recorded voice commands (Gábor et al., 2022) to prevent from giving inadvertent clues, while medical research is routinely conducted "double-blind" such that the experimenter does not know—and therefore cannot reveal—the ground truth of the experimental manipulation (Friedman et al., 2010). A fundamental in the avoidance of task-related confounders is the randomisation, variation and counterbalancing of stimuli.

In AI evaluation, it has not been standard practice to comprehensively vary and counterbalance stimuli within benchmarks, with examples of this approach only beginning to emerge in recent years (Rutar et al., 2024; Momennejad et al., 2023; Mizrahi et al., 2024; Wang & Zhao, 2024; Cao et al., 2024). The consequences of omitting this methodological practice have become evident through the results of numerous studies of RL agents and NLP systems performance (Krakovna et al., 2020; Gururangan et al., 2018; Poliak et al., 2018; Si et al., 2019; Sugawara et al., 2020; Kavumba et al., 2019; Niven & Kao, 2019). In both cases, systems have been shown to pick up simple surface-level cues that are predictive of the correct answers, allowing them to achieve high performance without using the information or capability putatively at test. Given the power and generality of modern large language models, it is reasonable to expect that these systems may have an even greater capacity to take advantage of such cues. This indeed has shown to be the case for some specific examples (e.g. Tu et al., 2020; Kavumba et al., 2022; Du et al., 2023).

In this paper, we take a broader approach to investigate whether a combination of such simple cues (in particular, unigrams and bigrams) can in principle be used to solve a wide selection of current LLM benchmarks: in practice, we test whether the ground truth answer labels can be predicted from these simple features of the prompt. We found that for a large number of benchmarks, simple classifiers were able to predict the ground truth with good accuracy.

In a fully controlled benchmark, it should not be possible to predict the correct answer from single or two-word or token features of the prompt. Agreement between the predicted label and the ground truth, as measured by Cohen's $\kappa$ (McHugh, 2012), should be close to zero. Traditionally, any value above 0.2 is taken to indicate a small but detectable agreement, anything above 0.4 fair-moderate and values above 0.6 considerable agreement (Landis, 1977). In our analysis, nearly half (9/19) of the benchmarks showed some evidence of agreement between predicted labels based on simple features, and the ground truth. Of these, most showed fair to moderate agreement, with some notable examples (e.g. Corporate Lobbying and SpaceNLI) showing consistent $\kappa$ values of above 0.6., indicating moderate to strong agreement.

In practice, researchers can repeat the analysis we carry out in this work during the dataset creation process; if they find significant predictive power with n-grams, they should identify which n-grams are most predictive (e.g., by examining coefficients in the logistic regression models) and introduce variations ensuring that ground truth labels are not correlated with those n-grams.

Our results results highlight a broader issue of bias in benchmark questions that allows for the use of alternative strategies to solve benchmark tasks: we focused on a relatively narrow set of simple cues (unigrams and bigrams), which is by no means a comprehensive overview of *all* the possible surface cues that could be used. Even if these specific cues are not present in a benchmark, others

may be present and possibly leveraged by tested LLMs. Overall, this suggests that care should be taken when designing a benchmark; for more information on how to do this, we refer to Rutar et al. (2024); Momennejad et al. (2023).

## 5.2 ARE LLMs USING SURFACE-LEVEL CUES TO SOLVE BENCHMARKS?

Our findings suggest that a system capable of recognising statistical regularities can achieve above-chance performance by relying solely on simple surface-level cues. This implies that LLMs may be exploiting these cues to pass benchmarks without demonstrating the intended capability. To investigate whether this is the case, we divided each benchmark into two subsets: instances where the classifiers predicted the correct answer and instances where they did not. By comparing LLM performance across these two groups, we aimed to investigate whether LLM scores were consistently higher for instances successfully predicted by the classifiers, suggesting that these LLMs might be using identified uni/bi-grams to solve benchmark tasks.

We observe that for certain model families, most notably those produced by OpenAI, Meta, and Mistral AI, there is a small but notable performance advantage in the predictable instances. Specifically, these model families perform better on instances where the classifier was able to predict the correct answer using unigram and bigram features. The fact that the effect is subtle suggests that these models are not relying solely on *these* surface cues, which is expected: even if the systems were entirely dependent on superficial patterns, it is unlikely they would restrict themselves to only cues from unigram and bigrams. Thus, the absence of evidence for reliance on $n$-grams in some model families does not imply that these models are not relying on any spurious features. Furthermore, many LLM families included only two models tested on just five benchmarks, making it more challenging to detect an effect of relying on superficial cues compared to, for example, the OpenAI family, which consisted of 18 models tested on at least 12 datasets (see Tables 3 and 4).

Conversely, the evidence we provide is not conclusive in determining that those LLM families rely on $n$-grams to succeed on the benchmarks considered. To do so, careful experimental manipulation—similar to those conducted in Gururangan et al. (2018); Niven & Kao (2019) and Kavumba et al. (2022)—would be necessary. Lacking these, confounding factors could explain our findings; for instance, it is possible that the instances successfully predicted by the $n$-gram models are coincidentally also easier, meaning that LLMs exhibiting the capability tested are more likely to succeed on those instances. While we do not believe this to be the case (indeed, as shown in Section 4.2, only a subset of LLMs shows a detectable pattern), our findings warrant further experimental manipulation.

In this work, we did not explore *how* LLMs may learn the statistical regularities present in benchmarks. Unlike traditional NLP models, LLMs are pre-trained and can be applied to different benchmarks out of the box (although techniques such as fine-tuning and few-shot prompting (Brown et al., 2020) can be applied). However, benchmark contamination, where an LLM is perhaps inadvertently trained on benchmark instances, is a widely-known issue (Achiam et al., 2023; Roberts et al., 2023; Jiang et al., 2024). Studies have also shown that fine-tuning LLMs on samples similar, but not identical, to benchmark tasks can improve their performance (Dominguez-Olmedo et al., 2024). Furthermore, the development pipeline for LLMs often involves supervised fine-tuning steps, which helps enhance their performance in question-answer settings. It has been suggested that this step may allow LLMs to learn to exploit spurious correlations (Du et al., 2023). We view the empirical analysis of existing LLMs to understand whether they rely on spurious features as complementary to the study of how they may learn to exploit such features. In particular, studying available LLMs may shed light on proprietary systems, for which analysing the development pipeline is not feasible.

## 6 CONCLUSION

We showed ways in which benchmarks could be solved without needing the assessed capability and provided evidence that LLMs may be using these surface cues to solve them. Future work should focus on identifying other types of cues that LLMs might be exploiting to solve benchmarks without demonstrating the intended capabilities being tested and extend our approach to open-ended question-answer datasets.

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

## A  LLMs TESTED ON THE CONSIDERED DATASETS

Table 4 reports the LLMs evaluated on the considered datasets. The results for the datasets belonging to LegalBench (Guha et al., 2023) were tested on multiple LLMs from different providers by the HELM project (Liang et al., 2022). We instead tested some of the other datasets on LLMs from OpenAI. Finally, the results for CLadder (Jin et al., 2023) and ProntoQA (Saparov & He, 2023) on some LLMs were available.

Figure 4 shows the distribution of accuracy (in terms of Cohen's $\kappa$, see Section 3.3) of the LLMs tested on each considered dataset.

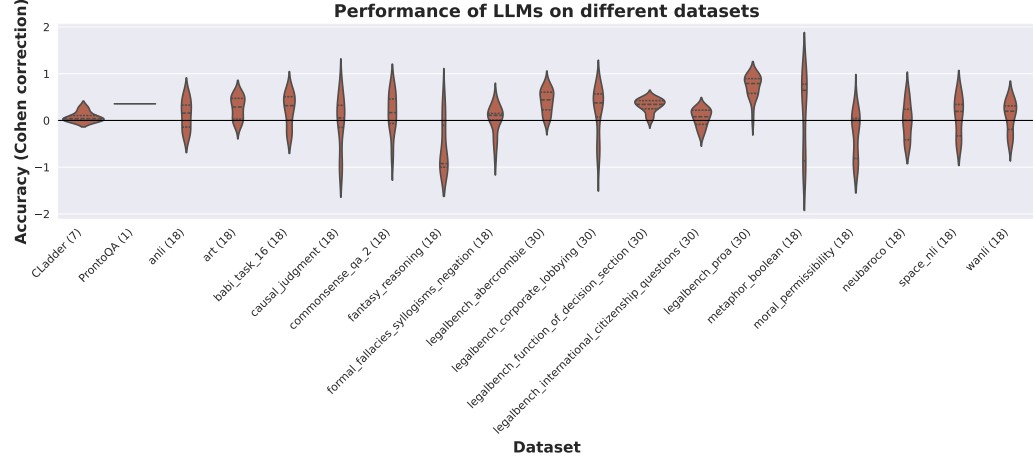

Figure 4: Distribution of accuracy (Cohen's $\kappa$) of the LLMs tested on each considered dataset.

| LLM | CLadder | ProntoQA | anli | art | babi_task_16 | causal_judgment | commonsense_qa_2 | fantasy_reasoning | formal_fallacies_syllogisms_negation | legalbench_abercrombie | legalbench_corporate_lobbying | legalbench_function_of_decision_section | legalbench_international_citizenship_questions | legalbench_proa | metaphor_boolean | moral_permissibility | neubaroco | space_nli | wanli |
|---|---|---|---|---|---|---|---|---|---|---|---|---|---|---|---|---|---|---|---|
| 01-ai/yi-34b | ✗ | ✗ | ✗ | ✗ | ✗ | ✗ | ✗ | ✗ | ✗ | ✓ | ✓ | ✓ | ✓ | ✓ | ✗ | ✗ | ✗ | ✗ | ✗ |
| 01-ai/yi-6b | ✗ | ✗ | ✗ | ✗ | ✗ | ✗ | ✗ | ✗ | ✗ | ✓ | ✓ | ✓ | ✓ | ✓ | ✗ | ✗ | ✗ | ✗ | ✗ |
| AlephAlpha/luminous-base | ✗ | ✗ | ✗ | ✗ | ✗ | ✗ | ✗ | ✗ | ✗ | ✓ | ✓ | ✓ | ✓ | ✓ | ✗ | ✗ | ✗ | ✗ | ✗ |
| AlephAlpha/luminous-extended | ✗ | ✗ | ✗ | ✗ | ✗ | ✗ | ✗ | ✗ | ✗ | ✓ | ✓ | ✓ | ✓ | ✓ | ✗ | ✗ | ✗ | ✗ | ✗ |
| AlephAlpha/luminous-supreme | ✗ | ✗ | ✗ | ✗ | ✗ | ✗ | ✗ | ✗ | ✗ | ✓ | ✓ | ✓ | ✓ | ✓ | ✗ | ✗ | ✗ | ✗ | ✗ |
| ai21/j2-grande | ✗ | ✗ | ✗ | ✗ | ✗ | ✗ | ✗ | ✗ | ✗ | ✓ | ✓ | ✓ | ✓ | ✓ | ✗ | ✗ | ✗ | ✗ | ✗ |
| ai21/j2-jumbo | ✗ | ✗ | ✗ | ✗ | ✗ | ✗ | ✗ | ✗ | ✗ | ✓ | ✓ | ✓ | ✓ | ✓ | ✗ | ✗ | ✗ | ✗ | ✗ |
| anthropic/claude-2.0 | ✗ | ✗ | ✗ | ✗ | ✗ | ✗ | ✗ | ✗ | ✗ | ✓ | ✓ | ✓ | ✓ | ✓ | ✗ | ✗ | ✗ | ✗ | ✗ |
| anthropic/claude-2.1 | ✗ | ✗ | ✗ | ✗ | ✗ | ✗ | ✗ | ✗ | ✗ | ✓ | ✓ | ✓ | ✓ | ✓ | ✗ | ✗ | ✗ | ✗ | ✗ |
| anthropic/claude-instant-1.2 | ✗ | ✗ | ✗ | ✗ | ✗ | ✗ | ✗ | ✗ | ✗ | ✓ | ✓ | ✓ | ✓ | ✓ | ✗ | ✗ | ✗ | ✗ | ✗ |
| anthropic/claude-v1.3 | ✗ | ✗ | ✗ | ✗ | ✗ | ✗ | ✗ | ✗ | ✗ | ✓ | ✓ | ✓ | ✓ | ✓ | ✗ | ✗ | ✗ | ✗ | ✗ |
| cohere/command | ✗ | ✗ | ✗ | ✗ | ✗ | ✗ | ✗ | ✗ | ✗ | ✓ | ✓ | ✓ | ✓ | ✓ | ✗ | ✗ | ✗ | ✗ | ✗ |
| cohere/command-light | ✗ | ✗ | ✗ | ✗ | ✗ | ✗ | ✗ | ✗ | ✗ | ✓ | ✓ | ✓ | ✓ | ✓ | ✗ | ✗ | ✗ | ✗ | ✗ |
| google/text-bison@001 | ✗ | ✗ | ✗ | ✗ | ✗ | ✗ | ✗ | ✗ | ✗ | ✓ | ✓ | ✓ | ✓ | ✓ | ✗ | ✗ | ✗ | ✗ | ✗ |
| google/text-unicorn@001 | ✗ | ✗ | ✗ | ✗ | ✗ | ✗ | ✗ | ✗ | ✗ | ✓ | ✓ | ✓ | ✓ | ✓ | ✗ | ✗ | ✗ | ✗ | ✗ |
| meta/llama-1-7b | ✓ | ✗ | ✗ | ✗ | ✗ | ✗ | ✗ | ✗ | ✗ | ✗ | ✗ | ✗ | ✗ | ✗ | ✗ | ✗ | ✗ | ✗ | ✗ |
| meta/llama-2-13b | ✗ | ✗ | ✗ | ✗ | ✗ | ✗ | ✗ | ✗ | ✗ | ✓ | ✓ | ✓ | ✓ | ✓ | ✗ | ✗ | ✗ | ✗ | ✗ |
| meta/llama-2-70b | ✗ | ✗ | ✗ | ✗ | ✗ | ✗ | ✗ | ✗ | ✗ | ✓ | ✓ | ✓ | ✓ | ✓ | ✗ | ✗ | ✗ | ✗ | ✗ |
| meta/llama-2-7b | ✗ | ✗ | ✗ | ✗ | ✗ | ✗ | ✗ | ✗ | ✗ | ✓ | ✓ | ✓ | ✓ | ✓ | ✗ | ✗ | ✗ | ✗ | ✗ |
| meta/llama-65b | ✗ | ✗ | ✗ | ✗ | ✗ | ✗ | ✗ | ✗ | ✗ | ✓ | ✓ | ✓ | ✓ | ✓ | ✗ | ✗ | ✗ | ✗ | ✗ |
| mistralai/mistral-7b-v0.1 | ✗ | ✗ | ✗ | ✗ | ✗ | ✗ | ✗ | ✗ | ✗ | ✓ | ✓ | ✓ | ✓ | ✓ | ✗ | ✗ | ✗ | ✗ | ✗ |
| mistralai/mixtral-8x7b-32kseqlen | ✗ | ✗ | ✗ | ✗ | ✗ | ✗ | ✗ | ✗ | ✗ | ✓ | ✓ | ✓ | ✓ | ✓ | ✗ | ✗ | ✗ | ✗ | ✗ |
| openai/ada | ✗ | ✗ | ✓ | ✓ | ✓ | ✓ | ✓ | ✓ | ✓ | ✗ | ✗ | ✗ | ✗ | ✗ | ✓ | ✓ | ✓ | ✓ | ✓ |
| openai/babbage | ✗ | ✗ | ✓ | ✓ | ✓ | ✓ | ✓ | ✓ | ✓ | ✗ | ✗ | ✗ | ✗ | ✗ | ✓ | ✓ | ✓ | ✓ | ✓ |
| openai/curie | ✗ | ✗ | ✓ | ✓ | ✓ | ✓ | ✓ | ✓ | ✓ | ✗ | ✗ | ✗ | ✗ | ✗ | ✓ | ✓ | ✓ | ✓ | ✓ |
| openai/davinci | ✓ | ✗ | ✓ | ✓ | ✓ | ✓ | ✓ | ✓ | ✓ | ✗ | ✗ | ✗ | ✗ | ✗ | ✓ | ✓ | ✓ | ✓ | ✓ |
| openai/gpt-3.5-turbo-0125 | ✗ | ✗ | ✓ | ✓ | ✓ | ✓ | ✓ | ✓ | ✓ | ✗ | ✗ | ✗ | ✗ | ✗ | ✓ | ✓ | ✓ | ✓ | ✓ |
| openai/gpt-3.5-turbo-0301 | ✗ | ✗ | ✓ | ✓ | ✓ | ✓ | ✓ | ✓ | ✓ | ✗ | ✗ | ✗ | ✗ | ✗ | ✓ | ✓ | ✓ | ✓ | ✓ |
| openai/gpt-3.5-turbo-0613 | ✓ | ✗ | ✓ | ✓ | ✓ | ✓ | ✓ | ✓ | ✓ | ✓ | ✓ | ✓ | ✓ | ✓ | ✓ | ✓ | ✓ | ✓ | ✓ |
| openai/gpt-3.5-turbo-1106 | ✗ | ✗ | ✓ | ✓ | ✓ | ✓ | ✓ | ✓ | ✓ | ✗ | ✗ | ✗ | ✗ | ✗ | ✓ | ✓ | ✓ | ✓ | ✓ |
| openai/gpt-4-0125-preview | ✗ | ✗ | ✓ | ✓ | ✓ | ✓ | ✓ | ✓ | ✓ | ✗ | ✗ | ✗ | ✗ | ✓ | ✓ | ✓ | ✓ | ✓ | ✓ |
| openai/gpt-4-0314 | ✗ | ✗ | ✓ | ✓ | ✓ | ✓ | ✓ | ✓ | ✓ | ✗ | ✗ | ✗ | ✗ | ✓ | ✓ | ✓ | ✓ | ✓ | ✓ |
| openai/gpt-4-0613 | ✗ | ✗ | ✓ | ✓ | ✓ | ✓ | ✓ | ✓ | ✓ | ✓ | ✓ | ✓ | ✓ | ✓ | ✓ | ✓ | ✓ | ✓ | ✓ |
| openai/gpt-4-1106-preview | ✓ | ✗ | ✓ | ✓ | ✓ | ✓ | ✓ | ✓ | ✓ | ✓ | ✓ | ✓ | ✓ | ✓ | ✓ | ✓ | ✓ | ✓ | ✓ |
| openai/text-ada-001 | ✗ | ✗ | ✓ | ✓ | ✓ | ✓ | ✓ | ✓ | ✓ | ✗ | ✗ | ✗ | ✗ | ✗ | ✓ | ✓ | ✓ | ✓ | ✓ |
| openai/text-babbage-001 | ✗ | ✗ | ✓ | ✓ | ✓ | ✓ | ✓ | ✓ | ✓ | ✗ | ✗ | ✗ | ✗ | ✗ | ✓ | ✓ | ✓ | ✓ | ✓ |
| openai/text-curie-001 | ✗ | ✗ | ✓ | ✓ | ✓ | ✓ | ✓ | ✓ | ✓ | ✗ | ✗ | ✗ | ✗ | ✗ | ✓ | ✓ | ✓ | ✓ | ✓ |
| openai/text-davinci-001 | ✓ | ✗ | ✓ | ✓ | ✓ | ✓ | ✓ | ✓ | ✓ | ✗ | ✗ | ✗ | ✗ | ✗ | ✓ | ✓ | ✓ | ✓ | ✓ |
| openai/text-davinci-002 | ✓ | ✗ | ✓ | ✓ | ✓ | ✓ | ✓ | ✓ | ✓ | ✓ | ✓ | ✓ | ✓ | ✓ | ✓ | ✓ | ✓ | ✓ | ✓ |
| openai/text-davinci-003 | ✓ | ✗ | ✓ | ✓ | ✓ | ✓ | ✓ | ✓ | ✓ | ✓ | ✓ | ✓ | ✓ | ✓ | ✓ | ✓ | ✓ | ✓ | ✓ |
| tiiuae/falcon-40b | ✗ | ✗ | ✗ | ✗ | ✗ | ✗ | ✗ | ✗ | ✗ | ✓ | ✓ | ✓ | ✓ | ✓ | ✗ | ✗ | ✗ | ✗ | ✗ |
| tiiuae/falcon-7b | ✗ | ✗ | ✗ | ✗ | ✗ | ✗ | ✗ | ✗ | ✗ | ✓ | ✓ | ✓ | ✓ | ✓ | ✗ | ✗ | ✗ | ✗ | ✗ |
| writer/palmyra-x-v2 | ✗ | ✗ | ✗ | ✗ | ✗ | ✗ | ✗ | ✗ | ✗ | ✓ | ✓ | ✓ | ✓ | ✓ | ✗ | ✗ | ✗ | ✗ | ✗ |
| writer/palmyra-x-v3 | ✗ | ✗ | ✗ | ✗ | ✗ | ✗ | ✗ | ✗ | ✗ | ✓ | ✓ | ✓ | ✓ | ✓ | ✗ | ✗ | ✗ | ✗ | ✗ |

Table 4: LLMs evaluated on each considered dataset.

