# OpenReview forum: "Leaving the barn door open for Clever Hans: Simple features predict LLM benchmark answers"
_ICLR.cc/2025/Conference — Submitted to ICLR 2025_

### Official Review · Reviewer_1HT7 · 2024-10-31

**Soundness:** 3
**Presentation:** 3
**Contribution:** 3
**Rating:** 5
**Confidence:** 2

**Summary:**

The paper examines multiple-choice benchmarks by training classifiers on various surface-level features, such as n-grams. This approach demonstrates that current evaluations may be susceptible to these spurious cues. The authors then use these classifiers to assess whether the performance of current models relies solely on surface-level features.

**Strengths:**

- The classifier approach presented in the papers is highly compelling for identifying whether spurious features (i.e n-grams features) exist in an evaluation.
- Meta-studies on evaluations offer an interesting and valuable research direction for the field.

**Weaknesses:**

- The results show that certain LLMs performance correlates with performance on spurious (e.g., n-gram) features. However, there could be other features the models rely on to achieve similar performance to these classifiers. To address this concern, it might be helpful to include case studies on these data points to gain a qualitative understanding of the specific behaviors. Such case studies/analyses would be particularly convincing in demonstrating that LLMs exhibit this behavior. As currently presented, the paper may benefit from additional evidence to clarify the extent to which these spurious features influence LLM performance. Although the authors even concede this point in section 5.2, for which I am grateful to the authors, I do think this evidence is required for a more compelling case for why the community should care about these types of spurious features in evaluation.
- The method due to its reliance on n-gram is limited to multiple-choice questions. It would be more compelling if different types of methods could be constructed for other types of evaluations, such as open-ended QA.

**Questions:**

- What was the criteria for the datasets chosen, it would be interesting to include MMLU?

---

> ### Author Response · Authors · 2024-11-19
>
> We thank the reviewer for their comments. Below are our responses.
>
> > ‘There could be other features the models rely on to achieve similar performance to these classifiers. To address this concert, it might be helpful to include case studies on these data points to gain a qualitative understanding of specific behaviours’
>
> We understand the concern that LLMs might rely on features beyond the n-gram patterns we identified. We explicitly acknowledge this in Section 5.2 and the introduction. The goal of our paper was to cast a wide net (44 LLMs, 11 model families, and 19 datasets) to identify areas or LLM families that may be exploiting the multiple n-gram features we considered. The goal wasn’t exhaustively identifying all possible features LLMs might leverage. We highlight that certain benchmarks can be solved using these simple features and that LLMs may be using these features to solve benchmarks. We believe our findings provide sufficient evidence to raise concerns about some of the current benchmarks used for LLM evaluation.
>
> Additionally, we emphasise that most previous works studying spurious correlations in LLMs (e.g., Gardner et al. (2021), Tu et al. (2020), Kavumba et al. (2022)) have focused on individual words or n-grams. In contrast, our work considers the combined effect of multiple n-grams simultaneously, thereby expanding the range of spurious correlations under consideration.
>
>
>
> > ‘The method due to its reliance on n-gram is limited to multiple-choice questions. It would be more compelling if different types of methods can be constructed for other types of evaluations, such as open-ended QA?’
>
>
> In this paper, we have chosen to focus only on multiple-choice benchmarks, as these constitute a substantial portion of commonly used benchmarks, and our method is readily applicable in this context. However, we note in the conclusion that future work should extend this method to open-ended QA, which would require substantial methodological innovation. Despite this limited scope and the use of simple features (n-grams) and predictive models (logistic regression), we find that many benchmarks widely used to assess LLM performance can be effectively solved using this straightforward approach.
>
>
> > ‘What was the criteria for the datasets chosen, it would be interesting to include MMLU?’
>
>
> We aimed to cover a wide range of linguistic and reasoning tasks. The benchmarks we chose evaluate causal reasoning, counterfactual analysis, moral judgement and decision-making, various types of formal reasoning, metaphor understanding, commonsense reasoning, spatial reasoning, legal reasoning, and natural language inference. For training logistic regression models, we focused on benchmarks that have consistent ground truth labels across all instances (e.g., all ground truth labels are ‘Yes’/‘No’ or ‘Correct’/‘Incorrect’). This led us to select 19 benchmarks. However, MMLU is a multiple-choice benchmark where the labels for each instance vary and was such unsuitable for our setting.

---

> > ### Comment · Reviewer_1HT7 · 2024-11-22
> >
> > I thank the authors for their response. While I find this paper intriguing and appreciate its insights, its contributions are limited without extending the framework to include additional types of evaluations beyond binary multiple-choice. Therefore, I am inclined to maintain my score.

---

### Official Review · Reviewer_VH76 · 2024-11-03

**Soundness:** 2
**Presentation:** 2
**Contribution:** 2
**Rating:** 5
**Confidence:** 4

**Summary:**

This paper focuses on exploring the "Clever Hans" effect, also known as the "shortcut learning" effect, in which the trained model exploits simple and superficial correlations instead of intended capabilities to solve the evaluation tasks. The authors demonstrate that simple classifiers (logistic regression) trained on unigrams and bigrams can potentially predict correct answers in several benchmarks. They also provide evidence suggesting that some LLMs may rely on these simple patterns to achieve high performance, potentially undermining the validity of the modern LLM benchmarks.

**Strengths:**

- The motivation is clear and interesting. Focusing on superficial correlations of existing benchmarks can help benchmark design and evaluation for LLMs.
- The methodology of using logistic regression to analyze n-gram features across multiple benchmarks provides a clear and easy way to evaluate the quality of test datasets.

**Weaknesses:**

- While the paper suggests that LLMs might be using n-gram patterns to solve benchmarks, it does not establish causality. It can be helpful to add experimental manipulation to directly test whether LLMs rely on n-grams.
- The findings may not apply universally across all LLMs or benchmarks, as the analysis is limited to specific datasets and model families. And based on the experimental results, only some families of LLMs demonstrate the possibility of relying on spurious correlations.
- The study focuses on a specific type of shortcut (n-grams) and may not capture other forms of spurious correlations that LLMs could exploit.

**Questions:**

Please see the weaknesses above.

---

> ### Author Response · Authors · 2024-11-19
>
> We thank the reviewer for their comments. Below we address their main concerns. We kindly ask the reviewer to revise their score if they feel our response adequately addresses their comments, or otherwise to indicate ways in which we can further address them and improve our paper.
>
> >  While the paper suggests that LLMs might be using n-gram patterns to solve benchmarks, it does not establish causality. It can be helpful to add experimental manipulation to directly test whether LLMs rely on n-grams.
>
> We explicitly acknowledge in Section 5.2 that our paper does not include experimental manipulation and, therefore, does not establish causality. Indeed, the fact that some n-grams are found to be predictive does not necessarily mean that an LLM will use them. However, an LLM could potentially use two or more disjoint sets of n-grams to enhance its predictions. Intervening on one set at a time might be ineffective if the model is still relying on other sets. As a result, performing a causal analysis is challenging without employing mechanistic interpretability. However, current techniques in interpretability are not sufficiently mature for this purpose.
>
>
> > The findings may not apply universally across all LLMs or benchmarks, as the analysis is limited to specific datasets and model families
>
>
> That is correct. However, our analysis includes 19 datasets and 44 LLMs across 11 model families, making it substantially comprehensive. While it is true that we could have included a larger number of models and datasets, this would have drastically increased the computational cost of our methodology.
>
>
> > Based on the experimental results, only some families of LLMs demonstrate the possibility of relying on spurious correlations.
>
>
> Indeed, we find that some LLM families demonstrate this behaviour, but we do not claim that others do not, as our methodology may not have identified the specific sets of n-grams those other LLMs rely on. Nevertheless, the fact that we find evidence that n-grams impact the performance of some LLMs suggests that this may also occur in other models. This highlights the need to address the root cause: building benchmarks where n-grams are not predictive of ground truth labels.
>
>
> > The study focuses on a specific type of shortcut (n-grams) and may not capture other forms of spurious correlations that LLMs could exploit.
>
>
> The reviewer is correct in pointing out that our analysis does not capture other possible forms of spurious correlations, as noted at the end of Section 5.1. Indeed, the fact that we find low label predictability for some datasets using n-grams (as shown in Figure 1) and observe no effect of n-gram predictability on the performance of some LLMs (Figure 3) should not be interpreted as evidence that no spurious correlations are present in those datasets or that those LLMs are not relying on them. However, we emphasise that most previous works studying spurious correlations in LLMs (e.g., Gardner et al. (2021), Tu et al. (2020), Kavumba et al. (2022)) have focused on individual words or n-grams. In contrast, our work considers the combined effect of multiple n-grams simultaneously, thereby expanding the range of spurious correlations under consideration.

---

### Official Review · Reviewer_PELq · 2024-11-04

**Soundness:** 2
**Presentation:** 2
**Contribution:** 2
**Rating:** 5
**Confidence:** 4

**Summary:**

This paper investigates an interesting question about the validity of benchmarks for existing AI research -- whether simple n-gram features can solve these benchmarks. They train logistic classifiers to extract these text features and use them to predict results on multi-choice questions. The experimental results suggest that many existing benchmarks leak with these n-gram features, and potentially limit the validity of benchmarks.
Furthermore, they test whether LLMs' predictions are based on these features. Their results show moderate evidence for the claim that models from OpenAI, Meta, Mistral AI, and Writer are more likely to provide correct answers for instances that n-gram models also successfully predictions.

**Strengths:**

The paper focuses on an important and timely issue of LLM evaluation -- whether the existing benchmarks are trivial and can be solved by simple text features, and furthermore whether LLMs rely on spurious cues to make correct predictions.
The motivation and the related work survey across different disciplines are good, providing a basis for future research.
The results of existing benchmarks that can be solved by simple text features are interesting.

**Weaknesses:**

The main weakness of this paper is the depth and comprehensiveness of empirical results.
The paper investigates an important question and attracts me in the introduction. However, the empirical results are disappointing. For example, the authors show that text features can solve these benchmarks. What's next? Which dataset is the easiest to solve or leak? What's your advice to improve these datasets? What happens if a more controlled dataset is used to test the same capability?

In addition, the authors need to provide more comprehensive results about how LLMs interact with these benchmarks.
Your experiments on LLMs' prediction show only moderate evidence that these features are used. Does it indicate that the leaked information is not used? If so, does it effectively lower the importance of this research?

Overall, I think the paper is not well-balanced with a big motivation and limited empirical results to support their motivation.

Missing reference:
Yan, Jianhao, et al. "Understanding In-Context Learning from Repetitions." The Twelfth International Conference on Learning Representations.

**Questions:**

If a dataset can be predicted by text features, does it mean that the tested capability can be bypassed?
Furthermore, how does a tested capability differ from identifying the word usage?

---

> ### Author Response · Authors · 2024-11-19
>
> We thank the reviewer for their comments. Below we address their concerns. We also updated our manuscript following some of the suggestions by the reviewer. We kindly ask the reviewer to revise their score if they feel our response adequately addresses their comments, or otherwise to indicate ways in which we can further address them and improve our paper.
>
> > Which datasets are easiest to solve?
>
> Figure 1 in our paper shows each dataset we considered, along with the accuracy of the best-performing classifiers built on simple features (using specific feature vectors). A Cohen’s kappa value greater than 0 indicates that the classifier performs above chance. Bolded datasets are those for which our simple logistic regression models achieved significant predictive power (i.e., kappa > 0.2). We found that there are 9 datasets where simple frequencies of 1-grams and 2-grams lead to kappa > 0.2. We clarified in the text how higher values of kappa indicate stronger predictive power.
>
> > What’s your advice to improve these datasets?
>
> Ideally, these simple n-gram features should not be predictive of the label. We advise researchers building benchmark datasets to perform this analysis during the dataset creation process. If they find that instances can be solved using simple n-grams, they should identify which n-grams are most predictive of the label (e.g., by examining the n-grams associated with the highest coefficients in logistic regression models) and ensure that ground truth labels are not correlated with those n-grams. For guidelines on building more robust datasets with higher validity, we point to [1]. We added these concrete suggestions to Section 5.1 of our paper, which discusses how spurious correlations can be avoided.
>
> > ‘The authors need to provide more comprehensive results about how LLMs interact with these benchmarks’
>
> Figure 2 in our paper shows how each LLM we evaluated performs (in terms of Cohen’s kappa) on the benchmarks selected for our analysis. We find that the majority of LLMs perform above the chance level on many benchmarks, although some perform at or below chance level. Additionally, the figure indicates that n-grams are more effective at solving benchmarks when LLM performance is above chance on those benchmarks and when their accuracy is higher on the instances that n-gram logistic regression models correctly predict.
>
> [1] Danaja Rutar, Lucy Gaia Cheke, Jose Hernandez-Orallo, Alva Markelius, and Wout Schellaert.
> General interaction battery: Simple object navigation and affordances (GIBSONA). Available at
> https://papers.ssrn.com/sol3/papers.cfm?abstract_id=4871025

---

> ### Author Response · Authors · 2024-11-19
>
> > ‘Your experiments on LLMs' prediction show only moderate evidence that these features are used. Does this indicate that the leaked information is not used? If so, does it effectively lower the importance of this research?’
>
>
> Our evidence is measured by the p-values obtained from paired t-tests testing the hypothesis that the average accuracy of a given LLM family is higher when logistic regression classifiers trained on n-grams correctly predict the label. This supports the hypothesis that some LLMs may be using n-gram features to make their predictions. However, we do not claim that the other LLMs do not use some kinds of shortcuts to solve problems, as our methodology may not have identified the specific set of n-grams those other LLMs rely on. Nevertheless, the fact that we find evidence that some LLMs may be using simple n-grams to solve benchmarks suggests that this may also apply to other models. This underscores the need to address the root cause: building benchmarks where n-grams are not predictive of ground truth labels, which is the key takeaway of our work. Thus, even if we find this effect only in a subset of LLMs, it does not diminish the significance of our research.
>
> > ‘If a dataset can be predicted by text features, does it mean that the tested capability can be bypassed?’
>
> Yes, this is correct. Since n-grams do not possess (nor encode in any way) the capability being tested, the fact that they can be used to solve the benchmark indicates that the benchmark can be solved without using the capability it is intended to test. We find 9 datasets (out of 19) where this is the case to some extent.
>
> > ‘How does a tested capability differ from identifying the word usage?’
>
> We are not entirely sure what the reviewer is referring to here. However, we would like to note that using a specific (cognitive) capability to solve a task is different from relying on simple n-grams to achieve the same outcome. For example, to succeed on the Fantasy Reasoning benchmark (part of BigBench), an LLM would need to reason about situations that deviate from real-world common sense. This involves inferring the rules of the fantasy world and making inferences based on those rules. This task cannot (and should not) be easily solvable by identifying simple 1-gram or 2-gram words or tokens. However, our findings show that logistic regression models based on simple 1- and 2-grams can solve this benchmark well above random chance. This suggests that there are specific 1- and 2-grams that an LLM could exploit to solve the benchmark without actually learning the rules of the fantasy world or making sophisticated inferences. As a result, we can conclude that the benchmark is not appropriate for evaluating fantasy reasoning.
>
> > Missing reference: Yan, Jianhao, et al. "Understanding In-Context Learning from Repetitions." The Twelfth International Conference on Learning Representations.
>
> We added this reference in Section 2.2

---

> > ### Comment · Reviewer_PELq · 2024-11-26
> >
> > Thanks for the authors' detailed response. Some of my concerns are addressed. I will raise my score to 5.

---

### Author Response · Authors · 2024-11-19

The reviewers all agree that the issue we address is relevant and important to improve the state of AI evaluation, and that our approach leads to interesting results regarding the quality of test datasets.

Condensing our responses to the reviewers, we emphasise the following points:
- We find that a large proportion of benchmarks (9/19) allow us to achieve significant predictive power for the ground truth label using simple n-gram features. We then show that some LLMs (of the 44 we considered, belonging to 11 families) perform better on the prompts where this can be done. However, the fact that we do not find evidence for other LLMs does not imply that they are definitely not relying on spurious features, such as other n-gram combinations that our classifier did not identify, or other kinds of features. Instead, the fact that we found an effect for some LLM families by fitting a simple classifier (logistic regression) is significant, as it suggests that this phenomenon could also occur in other LLMs. This highlights issues with benchmark constructions and LLM evaluation. It is therefore paramount that researchers are aware of this issue, as identified by our paper, so that benchmarks can be designed to mitigate it. Ultimately, it should be the responsibility of benchmark developers to ensure that benchmarks are valid and robust tests. Separate research endeavours, like ours, should not be expected to analyse all sets of benchmarks and identify all possible spurious features.
- Relatedly, our analysis focuses on a specific type of spurious features, namely, unigrams and bigrams. Previous works (e.g., Gardner et al. 2021, Tu et al. 2020, Kavumba et al. 2022) mostly studied the effect of individual unigrams or bigrams on LLM performance. In contrast, our classification approach considers the combined effect of multiple n-grams simultaneously, thereby significantly expanding the range of spurious correlations under consideration. As noted in Sections 1 and 5.2, other forms of spurious features may also be predictive of ground truth labels and leveraged by LLMs. Clearly, it is impossible to consider all possible features in a single work. However, we would like to stress again how the effect we find with unigrams and bigrams indicates that similar issues may arise with other features as well. Hence, our findings raise concerns about the current state of benchmarking, underscoring the importance of developing benchmarks robust to a wide range of spurious features.

---

### Meta-Review · Area_Chair_cSzb · 2024-12-23

**Metareview:**

This paper addresses an interesting topic in the evaluation of large language models: the internal validity of benchmarks. Specifically, the authors investigate whether benchmarks designed for LLMs can be solved using superficial patterns, such as n-grams. By training ngram based classifiers, they demonstrate that these classifiers can achieve high performance on several multiple-choice benchmarks. Furthermore, the authors provide moderate evidence that some LLMs might also exploit these superficial patterns, by comparing ngram based predictions with LLM's predictions.

Strengths of the Paper
1. The paper addresses an important and underexplored issue in LLM evaluation, the internal validity of datasets.
2. The use of logistic regression classifiers to identify spurious n-gram patterns is reasonable. It provides a straightforward and interpretable way to evaluate dataset quality.

Weaknesses of the Paper

While the assumption and experiments are interesting, I agree with the reviewers that findings and analysis of this work are also a bit 'superficial'. First the study does not clearly expand its impact to 'how to improve LLM benchmarks'. In the rebuttal the authors suggest "n-gram features should not be predictive of the label" is unclear and may not be completely correct cause I believe the ability to extract semantics from ngrams is also important to LLMs. So perhaps we should always consider ngram classifier as a baseline and presents the delta improvement of LLMs over the baselines? Generally, the paper would be much stronger if it can propose actionable, detailed methods to mitigate this issue.
Also, the study is limited to multiple-choice tasks and n-gram features, leaving out other benchmark types and other potential spurious features.

In conclusion, while the paper raises critical concerns and provides an interpretable methodology, its findings are relatively limited in depth and scope. The lack of actionable mitigations and narrow task/feature focus restrict its potential impact. I agree with the reviewers that this paper is marginally below the acceptance threshold.

**Additional Comments On Reviewer Discussion:**

1. Authors highlighted the datasets where n-grams were most predictive and provided concrete recommendations for improving benchmarks.
2. Authors acknowledged that the study does not establish causality and explained the challenges. They emphasized that the results raise valid concerns about benchmarks' internal validity, even if causality is not included.
3. Authors argued that their study included a broad range of datasets and models. They explained that MMLU was excluded due to inconsistent ground truth labels, which made it unsuitable for their logistic regression approach.

While the authors made meaningful improvements during the rebuttal period, the reviewers believed significant concerns remained, especially in the limited analysis depth, and scope of tasks/features.

---

### Decision · Program_Chairs · 2025-01-22

Reject